# School food and nutrition environments in Australian primary schools: A scoping review

Tina Gingell[1,2], Emma Esdaile[1,2], Danielle Gallegos[1,2*]

1 Centre for Childhood Nutrition Research, Queensland University of Technology (QUT), South Brisbane, Australia, 2 School of Exercise and Nutrition Sciences, Faculty of Health, Queensland University of Technology (QUT), Kelvin Grove, Australia

* Danielle.gallegos@qut.edu.au

## Abstract

Schools are a key environment which can provide access to food, education and skills to enhance food and nutrition literacy and health and wellbeing of primary school students. This review set out to identify key features of school food and nutrition environments (SFNEs) of Australian primary schools, the barriers and enablers for optimising these features, and the impact of school socio-economic status on these features. Six health databases were searched between August 2021 and October 2024. Articles which met the inclusion criteria (n = 148) were selected for a content analysis using a school-system approach. A quality assessment and an in-depth review was then conducted on articles published post 2019 (n = 71). The content analysis identified 18 SFNE elements which were present within the classroom (n = 1), the classroom but externally-operated (n = 1), the school (n = 4), the school but externally operated (n = 6), and other elements external to the school (n = 6). The in-depth analysis revealed a complex interrelationship between these elements and the actors within these elements (students, parents, school staff, volunteers, organisations, community members, and government). As societal and school expectations evolved, this impacted the SFNE. Numerous enablers and barriers to food and nutrition education, student food consumption, and government and school policies were identified, which could be leveraged to optimise SFNEs. Fewer than a quarter of the articles considered the effect of socio-economic factors on the identified SFNE elements. SFNEs are complex, made up of numerous overlapping structural and relational elements. At the local level, the SFNE of every school is unique and school stakeholders (students, parents, school leaders and teachers) should be centralised in the development of local, contextualised strategies to improve their SFNE. State and national resourcing needs to consider financial support as well as time/capacity, personnel, expertise, and curriculum embedment.

**Data availability statement:** Full details of the quality assessment and results of the content analysis files are available from the QUT Research Data Finder database https://doi.org/10.25912/RDF_1731631788320

**Funding:** This research was funded by the Queensland Children's Hospital Foundation philanthropic grant from Woolworths staff and customer donations. The funders played no role in the design, implementation or reporting of the research.

## Introduction

Childhood is a critical window for the development of life-long eating habits, positive relationships with bodies and food, and the development of food preferences which span across the life-course [1–4]. Internationally, schools are recognised as a key enabling environment to provide access to nutrition and food as well as education and skills for food and nutrition literacy to optimise growth, development, health and wellbeing [5–7]. For children in primary school where there is less autonomy and control over their food, school environments are therefore critical for developing food agency and the optimisation of dietary intake for health and wellbeing [8]. Schools are examples of contained food systems, where food is procured, grown, prepared, distributed, sold and waste is disposed of [9]. Understanding these environments including the resources available and the systems in place is imperative if school food and nutrition environments are to be optimised.

Several previous reviews have sought to explore, within the primary school setting, health and/or nutrition interventions [10–15], integration of nutrition education into the curriculum [16], the impact of nutrition policies on the SFNE [17,18] and the environmental impacts of lunchboxes [19]. However, to the best of the knowledge of the authors, no reviews have simultaneously explored all factors that affect the SFNE. This review set out to identify the key features of the internal and external SFNE in Australian primary schools. In addition, this review sought to investigate if these key features varied by the school's socio-economic status (based on the geographical location of the school) and to identify barriers and enablers that could be addressed to optimise the SFNE of Australian primary schools.

## Methods

### Context

This scoping review explores food and nutrition environments in Australian primary schools. The governance of education in Australia sits across two layers of government. The Commonwealth Government provides states with funding and writes the curriculum and syllabus for compulsory years of education. The Australian Curriculum is developed nationally and interpreted and implemented by state jurisdictions. State governments (in Australia, Queensland, New South Wales (NSW), Victoria, Tasmania, South Australia (SA), Western Australia (WA), Australian Capital Territory (ACT), and Northern Territory (NT)) write the curriculum and syllabus for the non-compulsory years of education, they administer schools and register and employ teachers. At the school level, the leadership team has some autonomy to undertake activities that align with state policy, funding and key performance indicator reporting. Children in Australia attend primary school from the age of four or five up until eleven or twelve before transitioning to high or secondary school. They attend for approximately six hours with options for out-of-school hours care.

Food and nutrition education (FNE) is one of twelve areas of focus within Health and Physical Education (HPE) in version 9 of the Australian Curriculum (from

Foundation to Year 10) [20]. FNE also features as context for learning in two additional learning areas, namely Technologies and Science [20]. FNE is also offered as a medium for learning within other curriculum areas (e.g., mathematics, English, Dance and Drama) although there are no indications of how widely this is undertaken [21]. At an informal level, learning in and around food and nutrition often occurs through incidental conversations that arise with students as they interact with the school food and nutrition environment (SFNE) [22].

Australia does not have a universal school meals program but rather food is predominantly brought from home in lunchboxes. Additionally, most schools also have canteens (tuckshops) where food and snacks can be purchased, however, they are not necessarily providing food every school day. In many schools, food is increasingly being provided through school breakfast programs (SBP). State government policy and standards govern school canteens, regarding the types of foods that can be sold and made available within school grounds. These standards are typically implemented using a nationally agreed "traffic light" system where foods that are not recommended are labelled as "red foods" (typically energy-dense and nutrient-poor foods and drinks), those where caution needs to be applied are "amber foods" (sometimes collectively referred to as "occasional" foods) and foods that are recommended by the Australian dietary guidelines are "green foods" (or alternatively called "everyday" foods) [23].

## Protocol and registration

This scoping review used the Preferred Reporting Items for Systematic reviews and Meta-Analyses extension for Scoping Reviews (PRISMA-ScR) checklist [24]. The protocol was registered on The Open Science Framework [25] and eligiblity criteria are provided in Table 1.

## Information sources

Six databases (APA PsycInfo, ERIC, Medline, CINAHL, Embase and Scopus) were systematically searched using the terms "primary school" AND "Australia" AND ("food" OR "nutrition") AND ("environment" OR "curriculum" OR "policy" OR "intervention"), and a combination of synonyms of these terms. Database functions were used to filter the results to include only peer-reviewed articles published in English from 2008. Searches were initially conducted in August 2021 and updated in March 2023 and October 2024.

## Search

S1 Appendix provides an example of the searches undertaken.

**Table 1. Inclusion and exclusion criteria for selection of articles.**

|  | Inclusion Criteria | Exclusion Criteria |
|---|---|---|
| Language | English | Not in English |
| Publication type | Peer-reviewed | Theses, media, magazines, books, editorials, conference, abstracts, study protocols |
| Publication year | 2008 to current | Pre-2008 |
| Study location | Australia |  |
| Study Setting | Primary/elementary schools, after-school care attached to primary schools (student age range between 4–12 years) | Early education and care (daycare); kindergarten or pre-school; middle or high school (adolescents); |
| Subject | Food and nutrition environment including curriculum, physical, digital and human assets, internal and external food and nutrition environment references | School management of specific illnesses relating to diet, e.g., diabetes or allergy. Curriculum areas not about food, nutrition, or eating. Food environments not directly referencing primary school contexts |

### Selection of sources of evidence

The results from the database searches were de-duplicated and title screened against the inclusion criteria by one author (TG) using EndNote [26]. Included articles were transferred to Rayyan [27] and underwent initial title and abstract screening by two authors (TG, EE) with the third author (DG) resolving any conflicts. Subsequent full text screening (with reasons for exclusion noted) was undertaken by a minimum of two authors (TG, EE, DG). A sub-sample of papers were screened prior to commencement to ensure inter-rater consensus. Disagreements were resolved by discussion.

### Data charting process and data items

Data were extracted from selected articles using Excel™ by one author (TG). It was not necessary to obtain additional data from investigators of selected articles. The following data items were extracted: authors; year of publication; study aims; study design; study location and timeframe; recruitment methods; participant details; data collection methods; and data analysis methods; study results; specific socio-economic results.

### Synthesis of results

A **content analysis** was completed on all selected articles to characterise the elements of the SFNE, as follows. Firstly, a list of a priori codes (SFNE elements) were developed based on the authors prior research into the school food system (not published). Selected articles were allocated to one or more codes. Additionally, new codes were inductively identified, and these were discussed and agreed by two authors (TG, EE) before being included in the code book. Descriptive analysis was conducted to explore the count of articles by code and socio-economic consideration.

Articles published since 2019 were selected for **in-depth review**. In 2019, the Council of Australian Governments Health and Education Councils jointly released the *Good Practice Guide* for supporting healthy food and drink choices in Australian schools [28], representing the first intergovernmental consensus of guidelines for school food provision. Both education and nutrition policy areas are dynamic and the last six years potentially captures changes to policy and practice that are relevant for contemporary considerations.

Allocation of codes for articles used in the preliminary content analysis were updated during the in-depth review where appropriate. Findings and results relevant for each code were extracted and synthesised narratively. Articles which reported on socio-economic factors (Socio-Economic Indexes for Areas (SEIFA Index), school location/remoteness) were also noted, data extracted, and analysed to independently highlight these factors in the results.

### Critical appraisal

A quality assessment was completed on all articles using the relevant Joanna Briggs Institute checklist [29], appraisals for quasi-experimental, cohort, case-study, cross-sectional and qualitative study types. Mixed methods studies were assessed using both the relevant quantitative and qualitative checklists. Each appraisal tool had between eight and eleven criteria. Articles which received affirmative responses for all applicable criteria were classified as high quality. Qualitative studies (including case studies) with three or more negative or unclear responses, and quantitative studies with more than half negative or unclear responses were deemed low quality. All remaining papers were designated as medium quality.

## Results

A total of 148 articles were selected from 2,937 records from six databases and one citation search. Fig 1 provides the PRISMA-ScR diagram which explains the selection process. There were 71 articles published since 2019 identified for the in-depth review.

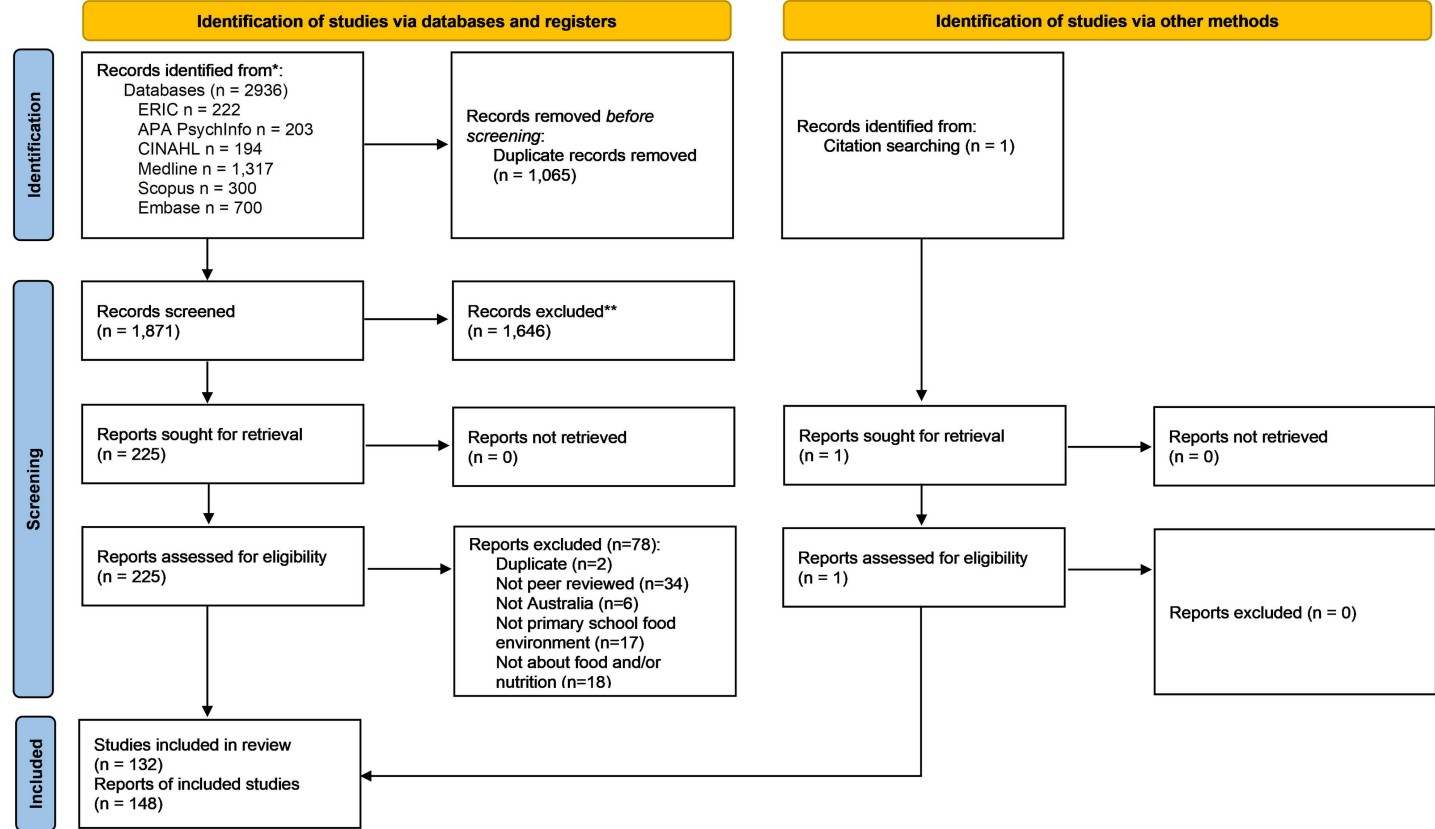

**PRISMA 2020 flow diagram for new systematic reviews which included searches of databases, registers and other sources**

*Consider, if feasible to do so, reporting the number of records identified from each database or register searched (rather than the total number across all databases/registers).

**If automation tools were used, indicate how many records were excluded by a human and how many were excluded by automation tools.

Source: Page MJ, et al. BMJ 2021;372:n71. doi: 10.1136/bmj.n71.

**Fig 1. PRISMA-ScR flow diagram.**

## Content analysis

In total there were 18 SFNE elements identified, and these were categorised into the following categories: classroom (n = 1), classroom-external (n = 1), school (n = 4), school-related external (n = 6) and external (n = 6). Classroom- and school-external elements were those that originated externally but were enacted within the classroom and school respectively. For example, some FNE programs were developed by external researchers and conducted as part of classroom lessons and were therefore deemed classroom-external elements. Another example is lunchboxes, which originate in the home environment but are consumed within the school environment and are therefore school-related external elements. Table 2 presents the results of the selected articles (n = 148) content analysis and SFNE elements and socio-economic considerations (full details can be found in Queensland University of Technology Research Data Finder [30]). Socio-economic considerations were reported in 23% (n = 34) of selected articles on average and were notably absent from education programs, resources, and lunchtime and other food breaks.

Overall, the literature on Australian SFNE touched on a broad range of elements. Although, some areas were reported more frequently, such as curriculum (classroom), education programs conducted in the classroom (classroom external

**Table 2. Content Analysis Results (*n* = 147).**

| SFNE Elements | *n*\* | *n* (%)\*\* articles with socio-economic considerations |
|---|---|---|
| **Classroom** | | |
| Curriculum | 14 | 1 (7) |
| **Classroom external** | | |
| Education program – classroom | 16 | 0 (0) |
| **School** | | |
| School food and nutrition policies | 10 | 1 (10) |
| Lunchtime and other food breaks | 6 | 0 (0) |
| Canteens | 22 | 5 (23) |
| Edible gardens | 16 | 3 (19) |
| **School-related external** | | |
| School breakfast programs | 15 | 4 (27) |
| Lunchbox contents | 15 | 5 (33) |
| Canteen online order system | 12 | 3 (25) |
| Outside of school hours care | 3 | 0 (0) |
| Food and nutrition communication with parents | 5 | 1 (20) |
| Canteen government policies | 18 | 4 (22) |
| **External** | | |
| Education programs targeting parents | 6 | 0 (0) |
| Government policies and initiatives | 14 | 4 (29) |
| Resources (including digital materials) | 9 | 0 (0) |
| Home food environment | 6 | 0 (0) |
| Billboards and advertisements | 9 | 6 (67) |
| Fast-food outlets | 2 | 2 (100) |
| **Total number of articles** | **148** | **34 (23)** |

SFNE – School Food and Nutrition Environment

\*Articles may be allocated to more than one code and therefore the total of this column does not sum to total (n = 147)

\*\*Percentages relate to corresponding denominator under the number of articles column

element), edible gardens, canteen (school element) and associated online ordering systems (external-school element), school breakfast programs, lunchbox contents and government canteen policies (school-external element), online resources and platforms, and other government policies (external element).

These results were used to create a visual representation of the Australian school food system, Fig 2. There is considerable cross-over with the elements, for example classroom curricular can interact with school elements such as the garden; school elements such as the canteen can interact with on the external-school element of the online ordering system.

### In-depth review of selected articles and synthesis

The in-depth review focused on articles published from 2019 (n = 71). A quality assessment was completed for each article, summarised in Table 3 (full details can be found in Queensland University of Technology Research Data Finder [30]). Although some papers were found to be of low quality, none were excluded from the in-depth review.

The categorisation of the SFNE for those articles that were reviewed in-depth is illustrated in Table 4, including the jurisdiction where the study was conducted. Based on the data extracted, each element is described and the barriers and enablers for implementation and relevant socio-economic considerations are synthesised. Most of the papers were based

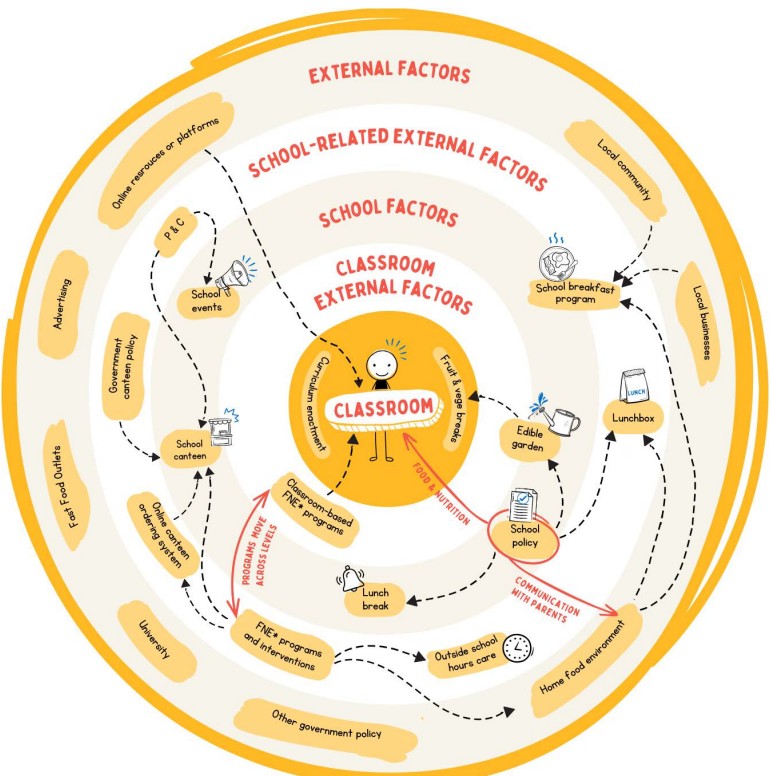

**Fig 2. Australian School Food System Map.** Legend: FNE – Food and Nutrition Education; P&C – Parents and Citizens.

**Table 3. Results of quality assessment of papers undergoing in-depth review (*n*=71).**

| Study Type | Total | High quality | Medium quality | Low quality |
|---|---|---|---|---|
| Quasi-experimental | 18 | 11 | 5 | 2 |
| Cohort | 2 | 0 | 2 | 0 |
| Cross-sectional | 26 | 15 | 9 | 2 |
| Qualitative | 18 | 2 | 8 | 8 |
| Narrative reviews/ case studies | 3 | 1 | 1 | 1 |
| Mixed methods | 4 | 0 | 1 | 3 |
| **Total** | **71** | **29** | **26** | **16** |

in NSW (44%), Victoria (18%) and WA (10%) with 17% of papers covering more than one state or were Australia wide. Queensland had only one paper, and no papers discussed school food environments in the NT or the ACT. Given that most publications were reporting on schools in NSW and the differences in how schools are supported to undertake food and nutrition work, comparisons across regional areas were not feasible.

## Classroom elements

The sole identified classroom factor was curriculum and the topics that integrated with food and nutrition. This SFNE element was also linked to other SFNE elements (discussed elsewhere), including nutrition education programs (classroom external factor), school policies, edible gardens (school elements), and online resources or platforms (external elements).

**Table 4. In-depth review of papers by element and jurisdictions.**

| SFNE Element | Count (*n*) | Jurisdiction | | | | | | |
|---|---|---|---|---|---|---|---|---|
| | | NSW | Vic | QLD | Tas | SA | WA | >1 state or Australia wide |
| **Classroom elements** | | | | | | | | |
| Classroom curricular | 12 | de Vlieger, 2019; de Vlieger, 2020 | Alston, 2019; Aydin, Booth, 2021; Aydin, Margerison, 2021; Aydin, 2022a; Bouterakos, 2020; Love, 2021 | | | Lalchandani, 2022; Velardo & Drummond, 2019 | | Aydin, 2022b; Follong, 2020 |
| **External-classroom elements** | | | | | | | | |
| Education programs in the classroom | 11 | Babic, 2023; Follong, 2022; Karpouzis, 2024; Poelman, 2019 | Connor, 2020 | Isbanner, 2022 | Kelly, 2022; Nash, 2020 | Velardo & Drummond, 2019 | | Poelman, 2020; Poelman, 2021 |
| **School elements** | | | | | | | | |
| School food and nutrition policies | 5 | | Alston, 2019; Aydin, Margerison, 2021; Love, 2021 | | | Lalchandani, 2022 | | Nanayakkara, Booth, 2024 |
| Lunchtime and other food breaks | 6 | | Aydin, Booth, 2021; Aydin, Margerison, 2021; Aydin, 2022a; Nanayakkara, 2024 | | | | | Aydin, 2024; Burton, 2022 |
| Canteen | 10 | Delaney, 2019; Wolfenden, 2019; Yoong, 2021 | Alston, 2019; Aydin, Margerison, 2021; Hill, 2023 | | | Lalchandani, 2022; Velardo & Drummond, 2019 | | Aydin et al, 2024; Nanayakkara, Booth, 2024 |
| Edible gardens | 5 | | Alston, 2019; Aydin, Booth, 2021 | Isbanner, 2022 | Holloway, 2023 | | | Rossi & Kirk, 2020 |
| **External-school elements** | | | | | | | | |
| Breakfast programs | 4 | | Alston, 2019 | | Jose, Vandenberg, 2020 | Watson, 2020 | | Jose, MacDonald, 2020 |
| Lunchbox contents | 12 | Manson, 2024; Reynolds, 2019; Sutherland, 2020 | Aydin, Margerison, 2021; Aydin, 2022c; Bouterakos, 2020; Love, 2021; Nanayakkara, 2024; Watson-Mackie, 2023 | | | Lalchandani, 2022 | | Aydin, 2022c; Aydin, 2024; Tanner, 2019 |
| Canteen online ordering systems | 10 | Beelen, 2021; Delaney, 2023; Leonard, 2021; Poelman, 2022; Stacey, 2021; Wyse, Delaney, Stacey, Lecathelinais, 2021; Wyse, Delaney, Stacey, Zoetemeyer, 2021; Wyse, 2019; Wyse, 2022 | | | | | | Billich, 2019 |
| Outside of school hours care (OSHC) and education programs in OSHC setting | 3 | Crowe, 2021; Crowe, 2022 | | | | | Forde, 2021 | |
| Food and nutrition communication with parents | 5 | de Vlieger, 2019; de Vlieger, 2020; Reynolds, 2019 | Aydin, 2022c | | | Lalchandani, 2022 | | |

*(Continued)*

**Table 4.** (Continued)

| SFNE Element | Count (*n*) | Jurisdiction | | | | | | |
|---|---|---|---|---|---|---|---|---|
| | | NSW | Vic | QLD | Tas | SA | WA | >1 state or Australia wide |
| Government canteen policies | 4 | Reilly, 2019; Wolfenden, 2019 | | | | | Myers, 2019; Pettigrew, 2019 | |
| **External elements** | | | | | | | | |
| Education programs targeting parents | 5 | Brown, 2021; Mihrshahi, 2019; Sutherland, 2021; Sutherland, 2019 | Marks, 2019 | | | | | |
| Other government policies and initiatives | 5 | Bravo, 2020; Innes-Hughes, 2019 | Brooks, 2019; Love, 2021 | | | | | Rosewarne, 2020 |
| Online resources and platforms | 6 | de Vlieger, 2019; de | Aydin, Margerison, 2021; Aydin, 2022a; Bouterakos, 2020; Love, 2021 | | | | Forde, 2021 | |
| Home food environment | 4 | | Aydin, Booth, 2021; Marks, 2019 | | | Velardo & Drummond, 2019 | | Tanner, 2019 |
| Billboards and advertisements | 4 | Richmond, 2020 | | | | | Parnell, 2019; Trapp, 2021; Wells, 2023 | |
| Fast-food outlets | 1 | | | | | | Trapp, 2022 | |

Legend: SFNE – School Food Nutrition Environment; NSW – New South Wales; Vic – Victoria; QLD – Queensland; Tas – Tasmania; SA – South Australia; WA – Western Australia

There were regional differences regarding expectations of who taught FNE and how to use the curriculum to teach FNE, for example, FNE was sometimes integrated into other curriculum areas [21,22,31]. An Australian-wide survey of primary school teachers found 60% (n = 61) of respondents integrated FNE with mathematics and HPE lessons [21]. In another survey, parents identified cross-curricular FNE opportunities in science, literacy, mathematics, and history [32]. Another study reported health and education government stakeholder support for cross-curricular integration of FNE into literacy and numeracy [31]. In Victoria, a curriculum review revealed that 2.6% of curriculum items included food and nutrition content, mainly within English, mathematics and humanities, followed by HPE, technologies, and science learning areas [31].

Nutrition topics taught in primary schools included cooking classes [22,33,34] and education on healthy choices [34,35]. Parents, teachers and students felt that lessons which included a practical component (for example, food diaries, group projects) were more effective at engaging their children [36,37]. Some examples of this include teacher use of food-related tools and equipment when teaching volume and capacity, such as household measurements, recipes, food and grocery store products, and to a lesser extent, food labels [21]. Teachers also taught about unhealthy choices, such as foods containing added salts, sugars, and saturated fats [35]. Some teachers also taught topics not included in the syllabus, such as eating behaviours, identifying healthy and unhealthy foods, and school gardens [35]. Finally, topics least discussed were reported to be portion sizes and macronutrients (carbohydrates, fats, and proteins) [35].

One Victorian study reported that teachers with a kitchen or garden more commonly incorporated FNE into their lesson plans compared to schools without these facilities [22]. Some parents noted their children were taught food and nutrition predominantly in the areas of environmental sustainability, and particularly for those with children attending a school with a

garden, growing, and preparing foods (gardening and cooking classes) [38]. Across several articles, most parents wanted more FNE embedded into the curriculum, particularly lessons aimed at improving cooking skills, gardening, sustainability, shopping, understanding food labels, and origins of food [32,37,38]. Topics which were not considered important by parents included food advertising, food safety, serving sizes, and nutritional composition of food [38].

Victorian teachers also discussed that events, special cultural days or classroom discussions during class teaching time and lunchtime provided opportunities to teach students about food and nutrition [22]. Some teachers also felt that a dedicated specialist teacher and supportive school management was necessary for FNE accuracy [36]. Teacher-reported experiences of FNE varied across studies and jurisdictions. Two NSW articles reported FNE was predominantly taught by teachers [35] or sometimes a guest speaker [35,37]. A small Victorian study found 25% (n = 5) of teachers did not include FNE in their lesson plans, citing this as a specialist area that was the responsibility of HPE teachers [31]. However, they also reported most of these specialist teachers focused on general health and wellbeing rather than nutrition specific topics [31].

Teachers and parents were divided about whether it was within the teacher's remit to teach FNE. In several studies, teachers reported teaching FNE as having a positive impact on children's food choices [22,31,36]. Parents who supported FNE in schools often noted the authority of schools, whereby their children listened to their teachers who were well placed to support healthy eating habits [32,36] and thus felt FNE should be taught by the school [38]. However, teachers also expressed caution around FNE, not wanting to appear judgemental or contradict parents' food choices, potentially creating conflicts with parents [31], for example, a family may have dietary habits relating to specific health concerns [39]. Some teachers felt FNE should be undertaken by parents and not teachers [35], and some parents felt children should learn about food at home and not during school [38]. These parents cited ongoing debates on what is considered healthy, and therefore what should be included/restricted in the curriculum [38].

Barriers to FNE included inadequate time and resources [22,35,36]; precedence given to higher priority topics such as English and mathematics [31,36]; inadequate support from principals; no availability of specialty or support staff for practical classes; high costs associated with cooking classes, gardening, and supermarket activities; inadequate nutrition education materials, school infrastructure and facilities; and a lack of nutrition knowledge by teachers and limited professional development in this area [22,35,36]. Other key stakeholders from health and education sectors (government, non-government organisations and academia) have acknowledged the difficulties associated with integration of FNE into the school curriculum [31].

Two studies explored student perspectives. Students identified key barriers to FNE including limited food and nutrition concepts, that were not scaffolded across year levels, lessons were not memorable, and they focused too much on individualistic health discourses [37,40].

**Socio-economic considerations.** No included articles reported results of socio-economic considerations relating to the Classroom SFNE element (including curriculum).

## Classroom external elements

This category included interventions that were developed by agencies and organisations external to the school but were implemented in the classroom. These programs used various approaches including: provision of lesson plans [41]; teacher training, leading to the development of lesson plans [42]; low and high intensity teacher training (via workshops) [43–45]; a whole-of-school approach incorporating components targeted at students (e.g., classroom education and fostering a positive eating environments), interpersonal relationships of students (e.g., teacher and parental involvement in the program), and community (e.g., providing resources for the canteen and breakfast programs) [46]; integration of nutrition education into other curriculum areas such as science or mathematics [47,48]; development of culturally rich learning "PITSTOPS" (two of five related to nutrition) [49]; and the creation of an artefact by students to be displayed during a school expo [50].

In evaluations of the efficacy of these programs students felt it may have improved their ability to select healthier alternatives [40,49] and parents from other program reported their children were requesting "healthier" food options at home [50]. Other aspects of education programs that were reported positively by parents and teachers included student engagement, motivation and autonomy over learning activities [50], engagement through food tastings [43], and having a hands on approach [41,43]. Some teachers and principals discussed resources in detail, and indicated it was important that the FNE resources (developed for these programs) align to the curriculum, are high quality, in digital format, and are comprehensive, including handouts for children and lesson plans that incorporate experience, and skills development [39,43].

Most teachers and principals involved in these programs felt it was important parents were involved in FNE. This was because they believed parents had a significant influence on the food choices of their children [39]. Some parents also felt their own food choices had a significant impact on their children's food intakes, and therefore wanted to learn more about nutrition [38] and some felt schools were well placed to provide information to parents on food and nutrition [36]. Some parents also expressed an interest in attending nutrition education session run by the school [51].

**Socio-economic considerations.** No selected articles reported the results of socio-economic considerations when implementing or evaluating nutrition education programs enacted in the classroom, the Classroom (external) SFNE element.

## School elements

School elements include the SFNE geographically located in school grounds. These include school nutrition policies, edible gardens which are sometimes utilised in the school curriculum, lunchtime and other breaks where foods are consumed, and school canteens where foods are purchased.

**School food and nutrition policies.** Jurisdictional differences in school food and nutrition policies were identified. An in-person school environment audit was undertaken in 2017 in two Victorian regional areas (northwestern and southeastern) and found most (55%) had some form of food/nutrition policy [33]. Conversely, a 2020 web-based audit in Greater Adelaide (SA) found no schools with publicly available nutrition policies or combined healthy eating and environmentally sustainable policies. However, in these schools, parents were encouraged to be part of a governing council, to support school activities, including children's health and well-being development [34]. In an Australia-wide survey, almost half (47%) of parents and teachers reported their school had a food-related rule or policy in place, such as avoidance of allergy foods, prohibition of the sharing of foods between students, and promotion of healthy eating [52]. Some studies also highlighted school policies relating to lunchboxes with respect to environmental and sustainable considerations [36,52], (see Lunchboxes section).

Some teachers felt a whole-of-school approach was necessary to ensure consistent food and nutrition messages were provided to students [31]. Some stakeholders discussed that a dedicated health promotion officer or school champions, such as a HPE teacher, was necessary to gain support throughout the school and community and apply a whole-of-school-strategy [31]. Furthermore, longer term initiatives require the support of parents, local communities and business partnerships to be successful [31].

**Lunchtime and other food breaks.** During school lunch breaks, many Australian schools have dedicated 'eating' and 'play' time. These eating times occur most commonly in an allocated outdoor area or in the classroom [53]. Teachers mentioned that incidental discussions sometimes occurred with students during these lunchbreaks around healthy eating practices, food waste and food culture [22], indicating this may be an important opportunity for FNE outside the curriculum. Teachers and parents recognised that teachers were an important role model to build healthy eating habits in students [36]. However, while most parents felt teachers should eat their lunch with the students, most teachers did not agree [53].

Issues were raised about the length of the lunch break [53–55] with the average time allocated being ten minutes or less [53]. More than half of parents and one-third of teachers believed this was inadequate, with longer lunchtimes associated with students consuming more food from their lunchboxes [53]. In addition to lunch breaks, many Australian primary

schools provide students with a 10-minute break during class time, usually in the morning, to have a fruit and vegetable snack (usually provided by the parents). Parents and teachers felt this additional break was an important time where students learnt about these everyday foods [22,36,38], and was an effective way to increase fruit intake [36].

**Canteens.** The 2017 Victorian regional school environment audit, mentioned previously, identified that 38% (n = 17) of school canteens offered a healthy canteen program [33]. Similarly, the aforementioned 2020 SA audit found some school canteen menus identified foods via the traffic light system and marketed healthy options via school newsletters [34]. Many schools have been found to deviate from government school nutrition policies by offering canteens specials for unhealthy foods during school celebrations and activities [34,40] or by not providing sufficient quantities of everyday foods [56]. Additionally, canteens menus generally offered occasional main meal items for sale cheaper than everyday main items [56]. Interventions targeting canteen managers that aimed to improve the adherence with state government policies have shown some success. Data from three randomised controlled trials were reanalysed to understand if 19 behaviour change techniques (as described by Michie, Richardson [57]) impacted school canteen's adherence to the NSW government's healthy canteen policy, *Fresh Tastes @ School* [58]. All behaviour change techniques were significantly associated with the removal of "red" foods and beverages from the menu and ensuring half the menu were "green" or everyday foods. One of these trials included a 24-month follow-up, post-implementation. It found intervention schools that met guidelines at 12 months continued to do so at 24 months and were more compliant post intervention than control schools [59].

Some teachers and parents felt that a healthy canteen menu (and therefore level of adherence to the state government policies) positively influenced food choices [36,54], while others felt that canteens contradicted the school's healthy food messages to students by supplying unhealthy foods [36,54,60]. However, some parents also viewed the canteen as a treat for their children and were not concerned about unhealthy food options available for sale [36,52].

**Edible gardens.** The 2017 Victorian regional school audit found nearly all schools (90%) in regional Victoria had a school vegetable garden [33]. The edible garden in a rural Tasmanian school was found to be a substantial aspect of the SFNE that fostered food literacy and potentially improved food security amount students and families [61]. Edible gardens enhanced student's knowledge and skills that led to improved behaviors around food production, nutrition and meal preparation, and ultimately better health and wellbeing [46,61]. For example, in regional Queensland, it was found students attending schools with a vegetable garden were more likely to know and recall the Australian Dietary Guidelines vegetable serves than students in schools without a vegetable garden [46]. One study reported that most Victorian schools were using the program developed by the Stephanie Alexander Kitchen Garden Foundation (SAKGF) [38]. This program was initially established to address childhood obesity by integrating health with the growing and preparing of foods in a school garden and then cooking and eating those foods [62]. It has since been adapted to align with the school curriculum potentially serving multiple purposes regarding specific curriculum areas, the promotion of healthy eating, and the enjoyment of new foods [38,62].

**Socio-economic considerations.** One study observed student purchases at the canteen and found students generally chose unhealthy options irrespective of school characteristics (e.g., school size, socio-economic status, remoteness) [63]. Another study found there were no difference in the price of cheapest main food item for sale at canteens across socio-economic advantage [56]. No other selected articles discussed the impact of socio-economic factors for any of the school SFNE elements.

## School-related external elements

External school elements were generated outside of the school or education system but implemented within the school environs.

**School Breakfast Programs (SBP).** The 2017 Victorian regional school audit [33] found 46% of schools in regional Victoria had a SBP for students (since this article SBP have been made universal in Victoria [64]). Although these programs are often established to address food insecurity [65], practically, students often chose to attend for many reasons, including

time constraints of their parents (juggling family and work commitments), having to leave home early or travel on the bus to school, and having access to different food options than at home [66]. Staff, volunteers, parents, and student attendees felt the benefits of a SBP went beyond providing additional food to alleviate hunger. For example, social interactions and relationship-building between students and with staff and volunteers was mentioned positively by all involved in these programs [65–67]. SBPs were also seen to improve student engagement in the school system; facilitated monitoring of well-being leading to early intervention when required, contributed to the development of manners, personal skills, and positive learning behaviours, provided adult role models, and improved academic performance [66,67]. Strengths of SBPs were strong community partnerships, and the satisfaction and pleasure gained by involved staff and key stakeholders who valued the program [65,66]. Challenges to providing a SBP included insufficient support and funding from government, schools, community partners and other key stakeholders, difficulties finding staff and volunteers to run the program, insufficient supplies (donated/rescued food) to support child preferences and the rapid expansion of the program as it moves beyond supporting food insecurity, and overreliance of key personal (often volunteers) to advocate for the program [66,67].

**Lunchboxes.** Schools attempted to manage lunchbox contents through methods such as lunchbox audits or supervisors [34,54], providing parents feedback on appropriate lunchbox foods, and lunchbox guidelines [54,68] to promote healthy eating among students. Information around food guidelines were communicated to parents through school websites, newsletters, enrolment packs, information sessions and school communication applications [34,54]. Parents in two surveys felt confident they had sufficient nutrition knowledge to pack a healthy lunchbox but faced many barriers [55,69]. This included the financial burden of purchasing healthier foods [55,69], especially foods which aligned to their children's food preferences (and therefore would be eaten and not wasted), which was of particular importance for students with special needs [69]. Additionally, some parents also faced issues around time for lunchbox preparation [55,69], particularly for parents who were employed [69]. Teachers felt that it was important to disseminate food and nutrition information to parents to ensure healthy foods were consumed by students [31].

Parents and teachers discussed the potential effects of lunchbox policies and guidelines on family households. Some parents felt environmental and sustainable lunchbox policies, such as "nude foods" policies, may have promoted healthy eating among students [36]. However, parents also discussed that lunchboxes can be a source of stress, anxiety and shame. Parents around Australia have reported teachers taking photos of student lunchboxes in order to communicate their beliefs around specific food items being 'good' or 'bad' [70]. This surveillance elicited great shame among the parents, particularly for those who prepared the lunchboxes photographed [54,70]. Other mothers who had not personally received feedback about their children's lunchboxes also felt lunchboxes were used to judge the quality of their parenting, including perceived judgement from teachers, other parents and self-judgement for not instilling healthy eating ideologies in their children [69]. Students were also negatively impacted, as they felt worry, embarrassment, fear and shame, and were exposed to social repercussions and reprimands from teachers and other students [54,70]. Lunchboxes were a key source of tension and potential conflict between teachers and parents [69,70].

**Canteen online order system.** Most student meals were ordered online where an online ordering system was available [71]. Foods sold via these online platforms were found to lack vegetables [72]. There were also differences in the cost of foods, with less healthy foods (assessed using the traffic light system) being generally cheaper than healthier foods [73]. However, when comparing between online and paper bag orders, no real difference in nutrition quality or content was identified [71].

Five cluster randomised controlled trials [74–79] and one randomised controlled trial [80] were implemented in online canteen ordering systems in NSW to improve the healthiness of lunch orders. These studies tested a variety of interventions, including providing tailored feedback into the order [74], menu labelling, repositioning of healthier items, prompting, feedback, and incentives to encourage healthier ordering [75–77], labelling foods according to the State's healthy canteen policy [78], adjusting menu item ingredients to include more vegetables [80], and incorporating consumer behaviour strategies into the display of the order [79].

Three of these studies found minimal or no changes to the purchases of everyday or "green" foods when comparing intervention and control schools [74,77,79]. However, two trials found significant improvements. In one intervention, schools reported increased vegetables purchases (77% increase) [80]. In the pilot of the second intervention, use of choice architecture strategies embedded in the on-line systems appeared to sustain the proportion of foods purchased by students in the intervention school categorised as everyday (<1% change) into the following term (compared to a 3% reduction in control schools) [78]. During the upscaled implementation, students attending the intervention schools had 7% more proportion of everyday food purchases (compared to a 3% reduction in control schools, i.e., 10% accumulated difference between groups) after 12 months [75], which increased to 10% at 18 months (compared to 6% increase in control schools, i.e., 4% accumulated difference between groups) [76]. These results show a narrowing of the differences between intervention and control group purchases of everyday foods overtime, which appear to be due to the control schools enacting changes in the absence of an intervention (potentially because of an observation effect), rather than the intervention schools losing ground. These changes also did not equate to large effects on energy density of lunch orders. During the pilot, there were no significant differences to total energy [78], while during the upscaled implementation, there were small but statistically significant differences in total energy (less than 300kJ) of the orders [75,76].

**OSHC and education programs in the OSHC setting.** During an audit of OSHC facilities in two health districts in NSW (n = 89), it was found that most (59%) had a nutrition policy in place, including 37% that were part of an umbrella organisation that had a nutrition policy [81]. OSHC facilities that were part of a larger organisation, were more likely to be using menu planning templates and checklists, and providing more vegetables, whole grains, and occasional (i.e., discretionary items) foods to the students than those that were smaller or independent [81].

Observations of OSHC food practices found that meals offered to students were generally in the form of fruit platters, sandwiches (with confectionary fillings such as jam, honey, cinnamon sugar, sprinkles/hundreds-and-thousand), and cooked meals (everyday and occasional foods) [81]. Foods mostly consisted of fruit, occasional foods, grains and dairy products [81,82]. Positive mealtime factors included staff sitting down with students, foods provided on platters, and the absence of sugar sweetened beverages [82]. Negative mealtime features were the occasional modelling of unhealthy food practices by staff (e.g., consuming occasional items) [82].

Many OSHC staff had never received formal food and nutrition training but nonetheless often felt confident to plan menus, locate nutritious meals and foods, and perform other nutrition related activities [83]. There were mixed outcomes on the effectiveness of nutrition related training provided to OSHC staff. One study [81] found training of staff made no impact to the foods offered to students, and another [83] reported staff felt improved confidence in menu planning, accessing food ideas, and delivering nutrition-based activities for children and this led to increased fruit and vegetable consumption among students. Additionally, this study specifically developed a Facebook™ group and a user-friendly website that created motivation among members and a supportive environment to share ideas, try out new health promotion activities and recipes which encouraged students to try new foods [83].

**Food and nutrition communication with parents.** In the web-based 2020 SA audit, over three-quarters (77%) of primary schools had healthy eating promotional resources in their curriculum, newsletters, or websites [234]. One school had also implemented a health advocacy group to address "junk food" concerns on school premises and which involved parents, teachers and students [34]. Other studies found schools communicated with parents about food and nutrition mainly through newsletters, but also through school apps, emails, brochures, face-to-face meetings, and workshops [35,51].

Online platforms were commonly used by schools as means for communicating about food and nutrition to parents [37,68]. Principals in one study felt that it was appropriate to provide healthy lunchbox messages to parents through digital communication methods at least monthly, and that it was suitable for these messages to come from a reputable third party [68]. The type of information provided to parents matters. One study found that while parents felt nutrition resources received from their school were not informative or useful, they were still interested in receiving more information from the school about nutrition [37].

**Canteen government policies.** Two studies [84,85] were conducted in WA and assessed school canteen compliance and motivation for compliance with the state government's canteen policy, *WA Healthy Food and Drink Policy* (WA Canteen Policy), ten years after its implementation. An audit of school canteen menus (n = 101 primary schools) found that, on average, menus included 72% "green items", 27% "amber items" and 1% "red items". This equated to 59% of primary schools meeting all traffic light targets (≥60% green items, ≤ 40% amber items and zero red items) [84]. Additionally, 89% of primary school menus met the green target, however, only 62% met the red target (details of amber compliance were not provided for primary schools). Most primary school menus also incorporated additional WA Canteen Policy requirements (e.g., fruit, plain water, and milk available, and the sale of amber savoury items restricted to ≥2 days per week) and almost half (47%) had colour-coded their menus [84]. No significant differences were found between schools located in metropolitan versus regional areas. In the second study, a survey among WA public school stakeholders (n = 307) – principals, teachers, canteen managers and Parent and Citizens (P&C) Committee presidents – a majority from primary schools (76%) found most stakeholders (84%) were motivated to meet the WA Canteen Policy and most desired (65%) to exceed policy requirements, particularly canteen managers (76%) [86]. Stakeholders were also asked their opinions on proposed policy extensions. The majority of respondents agreed with: a menu consistent with classroom health curriculum (83%); preservative/additive-free foods (78%); foods priced by healthiness (72%); healthy eating information included in school newsletters (70%); prioritising local foods/produce (68%) and seating areas for children (65%). Finally, most respondents were interested in additional resources to support policy implementation, such as information for parents, student assignments on the topic, healthy lunchbox, etc. [86].

The NSW *Fresh Tastes @ School* (NSW Canteen Policy) was assessed in one regional area of NSW in two studies [59,85]. The first study included a canteen intervention and found compliance with the NSW Canteen Policy increased in the control group from zero in 2013 to 24% in 2016 [59]. The second study found similar compliance levels (20%) with the NSW Canteen Policy in 2017 [85]. Canteen managers identified reasons for low compliance as the inability to influence student behaviour, inadequate skills, low self-confidence, lack of reinforcement, and poor ability to set/meet goals for the canteen [85].

**Socio-economic considerations.** Two studies considered the socio-economic impacts of school lunchbox contents and costs. One study (reported in two articles) [87,88] found that students attending schools located in disadvantaged areas were significantly more likely to have lunchboxes containing more serves of everyday foods (average 0.21 serves), more serves of occasional foods (average 0.4 serves), and total higher total energy (average 306kJ) [87], than students going to schools in advantaged areas. The energy sourced from everyday foods were similar regardless of living in advantage/disadvantage areas [87]. Additionally, the cost of foods contained within these lunchboxes (of students attending schools in disadvantaged areas) tended to be more expensive [88]. The second study supported these findings. A parent survey found there was no difference between level of disadvantage of the school and time spent by parents preparing lunchboxes or the cost of lunchbox foods [55]. However, parents with lower socio-economic status were significantly more likely to pack occasional foods (based on a scoring system) than parents with higher socio-economic status, but there was no difference in the amount of everyday foods [55].

Two studies [72,73] considered socio-economic factors when assessing online canteen ordering systems. One study found schools located in more advantaged areas offered more vegetables with hot meals (but not any other meal type) than schools in more disadvantaged areas [72]. Another study reported that schools in most disadvantaged areas tended to sell less healthy foods cheaper than healthy items [73].

Finally, one study reported that online platforms were less likely to be used in schools located in more disadvantaged areas as means for communicating about food and nutrition to parents [68].

### External elements

**Education programs (beyond the classroom).** Three lunchbox interventions aimed to improve the types of food included in children's lunchboxes [89–92]. These interventions used behaviour change models, such as the Behaviour

Change Wheel [91,92] and the Health Belief Model [89], to target parents' behaviour through improved nutrition knowledge. One intervention also included strategies to improve the school food environment (e.g., by supporting the development of nutrition policies) and children's knowledge (e.g., by providing curriculum resources) [91,92]. These programs were found to have positive impacts, including: establishment of lunchbox nutrition guidelines at intervention schools [92]; parents having more confidence and able to make healthier lunchbox choices after the intervention [90,92]; and significant improvements to parents' knowledge of dietary guidelines regarding fruits and vegetables [90].

The effectiveness of different types of information provided to parents was also assessed. Messages in the 'benefits', 'susceptibility', and 'severity' constructs of the Health Belief Model and those discussing health benefits (compared to discussing food groups) had a greater influence on a parent's intention to pack a healthy lunchbox, while personalisation of messages (e.g., by including the child's name) had no impact on parent's intentions [89]. Parents' acceptance of push notifications also diminished over time, reducing the effectiveness of ongoing intervention messaging [92]. Furthermore, these programs had minimal or no long-term impact on actual lunchbox contents [90–92].

Another article discussed the importance of relationships between schools and communities to ensure health promotion programs, such as obesity prevention, were effective [60]. School stakeholders (principals, canteen managers, school council committee members and teachers) interviewed felt that some parents were actively involved and supported these programs, but ultimately there was minimal action for health promotion by the community, for example, from local health departments, parents and schools [60].

**Government policies and initiatives.** An overarching review of state government nutrition policies across Australia published in 2020 found school nutrition policies for each state and territory, were mandatory for all public schools except in Tasmania [93]. All policies contained essential components for success, but some lacked monitoring and accountability components. For example, all school nutrition policies had implementation guides and tools, extras resources to support implementation, and except for SA, a guide for catering, fundraising, and advertising. They also had nutrition standards based on the *Australian Dietary Guidelines*. All state and territory nutrition policies, except Queensland and Tasmania, included roles and responsibilities for policy implementation and adherence. The ACT and NSW school nutrition policies included an outline of a monitoring and evaluation plan, and NSW, Queensland, and Tasmania included monitoring and evaluation tools. Tasmania, Victoria, and WA school nutrition policies also had an accreditation program [93].

In 2015, the NSW Healthy Children Initiative (HCI) was established to reduce childhood overweight and obesity by 5% by 2025 [94,95]. This included investing in an embedded health promotion workforce within regional health districts and upscaling several trialled programs to achieve this target. Within a suite of programs, the Live Life Well @ School (LLW@S) program was established in the primary school setting [94]. The LLW@S program included ten 'desirable school practices' across the areas of the curriculum, food and physical activity environment, professional development, and included extensive monitoring and reporting mechanisms [95]. By 2017, 83% of all primary schools in NSW were trained and participating in this program [94], and of those participating, most had achieved the progressively higher program targets in 2016 (79% of school had achieved 70% of the desirable practices, n = 1639) and 2017 (73% of schools had achieved 80% of desirable practices, n = 1495) [95].

One of the aspects of the HCI which supported success, were using programs with setting-based approaches that fostered relationships between policymakers, the local health promotion workforce and the schools in their district [94]. This approach created flexibility in the programs to meet the needs of the local community. It also upskilled partner organisations to incorporate health promotion activities into their daily practices. Furthermore, central monitoring of the programs ensured accountability and provided an avenue for feedback to program managers. Finally, it contributed to dialogue around childhood obesity and encouraged investment in health promotion [94].

In Victoria, the local government in Geelong had been supporting the local implementation of *The Achievement Program* since 2012, which was then scaled-up to a state-wide service. This program is similar to the LLW@S initiative,

except that program implementation is carried out by a not-for-profit organisation. *The Achievement Program* used a whole-of-setting approach to reduce childhood obesity and related chronic diseases in three settings, schools, childcare, and workplaces [31]. By 2015, 70% of primary schools in the City of Greater Geelong had registered for the program, of which 15% had achieved a "Health Promoting Setting" status [96]. Government-led initiatives were also supported by stakeholders who felt that state-based nutrition policies were important for fostering relationships between the health and education settings and provided leverage for funding school initiatives [31].

**Resources (including digital materials).** Teacher and staff accessed a variety of nutrition resources and platforms online, including from evidence-based websites, other websites, and Facebook™ groups [22,31,35,39,83]. These sites were mainly discovered through word-of-mouth and internet searches [31]. They also reported using Government resources, such as the *Australian Guide to Healthy Eating* [22], purchasing teaching resources from teaching suppliers, such as *MAPPEN*, *Twinkl* and *Teach Starter* [22], utilising resources provided through professional development courses, [17], and accessing resources from other places, such as food retailers [22]. Teachers also utilised nutrition program materials, such as resources provided through the *Stephanie Alexander Kitchen Garden Program* and *Cultivating Community Program*, and physical materials such as food and gardens during their lessons [35]. Teachers were reported to be continually interested in accessing credible new and updated resources [39] and felt that their personal beliefs around nutrition should not be relied upon [31].

Many teachers felt there were insufficient nutrition education materials, and those available often required updating [35,36,39]. Teachers and stakeholders (government, non-government organisations and academia) also discussed the usefulness of available resources. They felt the most useful nutrition resources were those which did not need a stable internet connection, were free or cheap and from a credible source, were fit-for-purpose, adaptable, aligned to curriculum, and were co-designed with education experts [31]. Lack of easy access to free online resources, concerns around cyber safety, insufficient classroom devices, unstable internet, and insufficient time to maintain digital devices/programs were reported as barriers to using nutrition resources in the classroom [31,39]. Furthermore, teachers discussed that a lack of reliable resources reduced their confidence to teach FNE to their students [36], and therefore may have contributed to less FNE being taught in primary schools.

**Home food environment.** The home environment can impact on the effectiveness of school programs. For example, school stakeholders (principals, canteen managers, school council committee members and teachers) in one school felt that the promotion of healthy eating (in the context of obesity prevention) by schools was less effective on students' food choices when parents had opinions or modelled behaviours which contradicted the program's messages [60]. The school environment may also influence the home environment. Some studies reported parents and students perceived children attempted to influence food choices in the home after learning about food and nutrition at school [38,40]. However, another study found that when the home environment is at odds with health eating messages taught to children by schools (particularly around a healthy lunchbox) parents may feel frustrated, concerned, worried, upset, and angry, and children can feel embarrassed, fearful, and ashamed [70].

**Billboards and advertisements.** Food and beverage advertising near schools was found to be widespread, including at food and non-food shops retailers, at the roadside, and on government infrastructure such as train stations and bus shelters [97–99]. Trapp and Hopper [99] found there were, on average, 25 (range: 0–190) food advertisements surrounding primary schools, and the presence of a shopping area near a school increased the presence of food advertisements by seven-fold. Furthermore, of these advertisements, the marketing power (assessed using an adolescent informed tool) of occasional food advertisements contained significantly more influential features compared to everyday food advertisements [100]. Richmond and Watson [98] found students were exposed to between two and ten food advertisements when travelling on train, bus or walking to school. Foods advertised were predominantly for occasional foods [97–99].

**Fast-food outlets.** One study investigated the occurrence of fast-food outlets surrounding schools located in metropolitan Perth [101]. They found almost half of primary schools (44%) had at least one fast-food outlet within 400

metres of the school grounds, over three quarters (77%) had one within 800 metres and most (86%) had one within one kilometre [101].

**Socio-economic considerations.** Government initiatives may have a good reach for specific "at risk" populations. Two studies assessed the uptake of government policies and initiatives in less disadvantaged areas. In the NSW LLW@S initiative, 71% (n = 432) of participating schools with a higher proportion of First Nations students, 72% (n = 769) of participating schools located in more disadvantaged area, and 65% (n = 28) of participating schools in remote areas had met 80% of the desirable practices in 2017 [95]. In Geelong (Victoria), there was a greater participation in *The Achievement Program* by schools located in the most disadvantaged quartile compared with those in the least disadvantaged quartile (78% v 52%), and this was also seen for completion of program objectives (22% vs 17%) [96]. However, it should be noted these rates were not linear across the socio-economic disadvantage quartiles [96]. Desirable practices in the areas of curriculum and food and physical activity environment were well adopted particularly when supported by other programs, such as *Crunch&Sip*. Professional development, monitoring and reporting practices were less adopted, particularly in areas where it may be difficult for schools to access appropriate training (e.g., remote locations) [95,96]. These findings suggest that schools in less disadvantaged areas may find these programs more beneficial for improving the SFNE for their students.Students attending schools located in lower socio-economic areas were exposed to significantly more unhealthy food and alcohol advertisements (within 250 metres of the school) [99] and which were more likely to contain more influential features aimed at children and adolescents [100], compared to schools located in higher socio-economic areas.

Finally, schools (primary, secondary and K-12 combined) located in low socio-economic areas were more likely to have a higher incidence of fast-food outlets within 400 metres of the school grounds compared to schools located in high socio-economic areas [101].

No other papers discussed the impact of socio-economic factors associated with external elements.

## Discussion

To the best of the authors knowledge, this is the first review which synthesises the impacts of internal and external factors on the SFNE, advancing a holistic approach to the study of SFNE in the literature. This review identified that within the Australia primary school system, classroom and other school environment factors played an important role in SFNE. Furthermore, factors external to the school interacted with aspects of the school system to influence the SFNE of Australian primary schools. For example, nutrition education programs designed by external organisations or research bodies often aimed to improve the skills and knowledge of student by creating programs enacted within the classroom [40,42–48,50]; while other interventions targeting canteen managers aimed to improve the quality of foods sold at the school canteen [74–80]. There were also external factors which did not interact with the school, but nonetheless impacted on the SFNE, such as interventions targeting parents (that is, the home environment) which aimed to improve the quality of foods packed into lunchboxes [89–92]. There were several elements of SFNEs that had been hypothesised as important by the authors of this review but were only mentioned briefly or not at all in the selected studies. These included school fund-raising, school events (e.g., school fetes and carnivals), school camps, relationships with local government, local government initiatives, regional farming, and external sporting/physical activity organisations (e.g., local swimming pools). This review identified the core elements that contribute to the complexity of SFNEs. These elements and relationships interact to affect how students engage with food environments and ultimately influence the development of their food-related behaviours.

Enablers and barriers for optimising SFNE were also identified in this review, broadly fitting into three categories: FNE, student food consumption, and government and school policies. The enablers for FNE in schools included: the school having a kitchen and/or garden [22]; special events or cultural days [22]; informal lunchtime conversations [22]; dedicated specialist teacher and supportive school management [36]; educations programs which included curriculum aligned

resources that incorporated fun activities that improved student engagement, motivation, student autonomy and involved parent [39,41,43,50,60]; integration practical and engaging nutrition lessons [36,37] and access to free/cheap, online, credible resources that are co-designed with experts and align to the Australian curriculum [31]. The main barriers to FNE included: inadequate teacher time and training [22,35,36]; the lack of prioritisation and inadequate support for embedding FNE in the curriculum [22,31,35,36]; a lack of school infrastructure and facilities to carry out nutrition lessons [22,35,36]; and insufficient access to free, ready-to-use, inclusive, and credible resources, and/or a stable internet network and sufficient classroom devices to utilise these resources [31,36,37,39,40].

Enablers to students consuming "green" foods included: Government canteen policies [84–86], support resources [86] and interventions [36,54,59,75,76,78,80,85] which aim to improve adherence to these policies; provision of food and nutrition information to parents about healthy lunchboxes [37,68,83]; and the modelling of healthy eating habits by parents, teachers and OSHC staff, and as well as eating food with students [36,82]. One significant barrier to this was identified – when the goals, values and messages of school policies and interventions conflict with parents' values. For example, lunchbox policies and health promotion interventions (e.g., obesity prevention) sometimes created stress and shame at home [54,69,70]. Additionally, parents modelling behaviours contrary to what was being taught, was seen negatively by the school [60]. Other barriers related to canteen managers (insufficient skills and confidence to effect canteen menu changes) [85] and financial and time constraints of parents when packing lunchboxes [55,69].

School and government policies were key enablers for improving SFNE. This review found that government and school policies and systems which embrace a whole-of-school and school-systems approach (e.g., LLW@S and HCI in NSW and *The Achievement Program* in Victoria) [31,94–96]; school resources (such as the presence of a school garden, offering a SBP, and sufficient funding) and schedules (such as sufficient break time for students to consume meals) that support SFNE [22,31,36,38,46,61,62,65–67].; and adequate nutrition training provided to staff [83].

The final aim of this scoping review was to explore the impact of socio-economic factors. The Australian food environment has been found to vary depending on the remoteness and level of disadvantage of the local area [102–105], indicating a likely similar impact on SFNEs based on geographic location and level of disadvantage. However, only a few of the selected studies considered the impact of a school's socio-economic area. However, only a few of the selected studies considered these factors. Only 23% (n = 34) of all articles and 23% (n = 16) [33,56,63,68,72,80,85,87,88,95–101] of articles included in the in-depth analysis discussed how socio-economic status affected their findings. This review found that schools in more disadvantaged areas were more likely to offer less healthy foods at lower cost than healthy foods [73] and less likely to use online platforms when communicating with parents about food and nutrition [68]. This indicates there may be a lack of resources in more disadvantaged areas that need to be considered when implementing school food systems change. Studies investigating the Australian food environment have similar findings and have identified volatile changes in food supply chains which impact the availability of foods in remote locations (which are often more disadvantaged socially also) [103,105]. These studies noted a reduced quality of foods in more disadvantaged neighborhoods [102,104]. More research is required to understand the impact of socio-economic factors on the SFNE.

### Further considerations

The term "healthy eating" is emotive, and when food ideologies do not align across different environments it may create additional pressures for families. Some parents felt their children's lunchbox contents were used to judge the quality of their parenting, including perceived judgement from teachers, other parents and mothers judging themselves for not instilling healthy eating ideologies in their children [69]. When enforced (e.g., through lunchbox audits), lunchbox policies can elicit shame among the parents, particularly for those who prepared lunchboxes identified as not healthy [70]. Students can also be negatively impacted, and feelings of worry, embarrassment, fear and shame can be created, and they can be

exposed to social repercussions and reprimanding from teachers and other students [70]. In one study [69], mothers interviewed had not personally received messages from the school regarding their children's lunchbox choices, however, they were worried nonetheless as they had heard about schools contacting other mothers. This indicates the effect of school lunchbox polices which conflict with the ideologies of home environments may be far reaching and should be implemented with caution.

School policies may also inadvertently promote an improper binary view of foods. For example, lunchbox audits that results in identification of foods as either 'good' or 'bad', or 'healthy' or 'unhealthy', relies on a narrow biomedical understanding of food that fails to incorporate cultural aspects of foods and ignores household barriers (e.g., financial and time constraints) [70]. The feeling of shame and failure invoked in parents and children from enforcing these lunchbox policies likely detract from the healthy food messaging goal of the school [70]. Furthermore, specific groups of students may have been marginalised by having more 'bad' foods, such as students from some cultural groups (e.g., who consume white rice) or experiencing greater disadvantage (and are unable to afford 'healthy' foods). Finally, some studies reported parents and students perceived that children attempt to influence food choices in the home after learning about food and nutrition at school [38,40], which may contradict parent's beliefs around healthy food relationships and therefore cause conflict. These differing food ideologies between SFNE factors suggest there is a need to ensure schools promote foods in a non-binary and inclusive manner.

This review also found the SFNE was impacted by the perceived role the school plays within the community, which may be evolving through societal changes such as family dynamics, living arrangements and work commitments, that sees schools potentially take on non-traditional responsibilities [65]. For example, SBPs in some schools have moved beyond their initial goal of providing food to those in need, to offering breakfast for all students [66]. Parents generally supported this realignment of scope. They felt schools had a responsibility to provide for community needs and a SBP portrays commitment and investment in the health and well-being of the community [65]. However, the school's responsibility for canteen provisioning and food monitoring is more complex. This can be seen through differing perspectives on schools providing healthy canteen menus, lunchbox policies and healthy eating education (as opposed to FNE).

### Implications for policy makers and researchers

Australian school food environments are a complex mix of multiple elements with diverse initiatives that vary across jurisdictions (states), schools and at the individual teacher level. The different approaches are positive as they highlight local adaptation and implementation but raises questions regarding structural support for local translation of broader policy. State-wide government initiatives however may provide an opportunity to improve SFNEs at a greater reach than locally created programs. NSW and Victorian policies provide two examples of this, noting that both these jurisdictions leveraged Commonwealth government funding to enable these policies. The investment by state governments into these initiatives further enabled health promotion activities into the community more broadly [80]. While the intergovernmental *Good Practice Guide* was published five years ago, we are yet to see if they have had the desired impact on school food provision. The far-reaching impacts of COVID-19 on schools is likely to have impeded these efforts. A renewed focus is needed to continue in this policy space. These examples highlight the need for stronger national co-ordination and funding to ensure states can more equitably support local initiatives to optimise SFNEs.

The approach of this scoping review has some utility for global applications. Due to the Australian context, findings here cannot be directly extrapolated to other nations. However, the holistic approach undertaken to conceptualise SFNE is a worthy endeavour for SFNE researchers and policy makers alike internationally.

A key implication of this review for researchers highlights the need to consider the interconnected nature of SFNE elements when conducting research, including reporting on socio-economic considerations and regional disparities. The use of an overarching framework (e.g., the Food and Agriculture Organisation *School Food and Nutrition Framework* [106], adapted for use in Australia) to reorientate research and interventions into SFNEs is warranted. Such a framework could

readily be used to inform policy in a consistent way, despite differences across Australian jurisdictions and the unique SFNE in every school.

SFNE across Australia have similar components, of which one is FNE. There are some barriers to implementation which would benefit from a targeted approach to building FNE interventions across curricular. Local implementation of initiatives to optimise SFNEs require a whole-of-school approach involving all stakeholders to develop and implement initiatives that aim to influence SFNEs with sustainability in mind. A whole-of-school approach, however, requires strategic planning and funding at national, state and regional levels.

## Limitations

Two main limitations of this review are worth noting. Firstly, while there is national oversight and co-funding of education in Australia, state governments are responsible for its provision. Most papers in this review emerged from NSW and Victoria. This could be due in part to the different approaches and priorities to health promotion across states, or that these two jurisdictions have invested significantly more in SFNEs than others. Structurally, NSW has an embedded health promotion workforce within regional health districts, and Victoria outsourced program delivery to not-for-profit organisations [107].

The second limitation was the limited reporting of analysis of the impacts of geographical and socio-ecological considerations on findings among the selected studies. Most publications came from NSW and very few studies considered the socio-ecological setting. If care is taken to interpret for local contexts, the findings of this study are valuable for the development of comprehensive strategies to improve SFNEs outside of NSW. However, strong recommendations cannot be made on socio-ecological aspects, except to note the need for researchers to collect data and report more broadly on these factors in research on SFNE.

## Conclusions

Australian SFNEs represent a microcosm of food systems in general. This scoping review identified their structural and relational complexity. Structurally, SFNEs are made up of interacting components across classrooms, schools, homes, other external areas, and government organisations. Relationally, SFNEs are inhabited by students, parents, teachers, school leaders and administrators, and canteen staff (including volunteers). While every school has a SFNE made up of these components, every school's SFNE is unique. More research and reporting of the implications of socio-economic and geographical disadvantage on SFNEs is needed.

Programs with flexibility to provide contextually relevant local support, delivered by health promotion officers with strong and ongoing relationships, are best-placed to support whole-of-school approaches to improving SFNE. Co-designing these approaches with students, parents, educators, and school stakeholders is more likely to yield embedded structural and cultural changes for the long term.

## Supporting information

**S1 Check list.**   PRISMA_2020_checklist for Paper.
(DOCX)

**S1 Appendix.  Example of the searches undertaken.**
(DOCX)

## Author contributions

**Conceptualization:** Tina Gingell, Emma Esdaile, Danielle Gallegos.

**Formal analysis:** Tina Gingell, Emma Esdaile, Danielle Gallegos.

**Investigation:** Tina Gingell, Emma Esdaile, Danielle Gallegos.

**Project administration:** Tina Gingell, Emma Esdaile, Danielle Gallegos.

**Supervision:** Danielle Gallegos.

**Visualization:** Tina Gingell, Danielle Gallegos.

**Writing – original draft:** Tina Gingell, Emma Esdaile, Danielle Gallegos.

**Writing – review & editing:** Tina Gingell, Emma Esdaile, Danielle Gallegos.

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
