## [Decision Letter · Decision Letter 0]

Dear Dr. Gallegos, 

Thank you for submitting your manuscript to PLOS ONE. After careful consideration, we feel that it has merit but does not fully meet PLOS ONE’s publication criteria as it currently stands. Therefore, we invite you to submit a revised version of the manuscript that addresses the points raised during the review process.

We look forward to receiving your revised manuscript.

Kind regards,

Mc Rollyn Daquiado Vallespin

Academic Editor

PLOS ONE

Journal Requirements:

2. We note you have included a table to which you do not refer in the text of your manuscript. Please ensure that you refer to Table 1 in your text; if accepted, production will need this reference to link the reader to the Table.

**Additional Editor Comments:**

Below, I have outlined the revisions needed:

**1. Introduction**

Refine the introductory section to ensure clarity and provide a comprehensive understanding of the study's scope.Emphasize the study's importance for child development from physical and mental perspectives.Clearly outline the unique contribution of the research to existing literature.

**2. Methodology**

Expand on the research methodology, ensuring transparency in design and execution.Include a detailed explanation of the quality assessment tool, particularly distinctions between high, medium, and low quality levels.Justify the use of the PRISMA model and ensure its implementation is clearly articulated.Address the influence of socio-economic factors on the methodology.

**3. Results**

Supplement the results with improved visual representations, such as diagrams, tables, or charts.Add clear captions and detailed explanations for all tables and figures, ensuring that abbreviations are defined where necessary.Discuss regional disparities in findings and their implications.

**4. Discussion**

Expand the discussion to include an in-depth analysis of results, mechanisms, and implications.Address the limitations of the study, including socio-economic factors and regional variations.Clarify the innovation and contributions of the study in advancing food and nutrition education.Discuss potential global applications or comparisons to other regions beyond Australia.

**5. Recommendations and Future Research**

Provide actionable recommendations for improving food and nutrition education in schools.Suggest future research directions to address gaps identified in this study, such as the socio-economic and regional disparities.

**6. Technical Revisions**

Revise for technical errors, including consistent formatting and explanation of abbreviations.Ensure that references are standardized and adhere to the required format.

Reviewers' comments:

Reviewer's Responses to Questions

**Comments to the Author**

1. Is the manuscript technically sound, and do the data support the conclusions?

Reviewer #1: Yes

Reviewer #2: Yes

Reviewer #3: Yes

2. Has the statistical analysis been performed appropriately and rigorously?

Reviewer #1: Yes

Reviewer #2: Yes

Reviewer #3: N/A

3. Have the authors made all data underlying the findings in their manuscript fully available?

Reviewer #1: Yes

Reviewer #2: Yes

Reviewer #3: Yes

4. Is the manuscript presented in an intelligible fashion and written in standard English?

Reviewer #1: Yes

Reviewer #2: Yes

Reviewer #3: Yes

Reviewer #1: Suggestion 1: Strengthen the Elaboration of Research Methodology

Suggestion 2: Augment the Explanation of the Quality Assessment Tool

Suggestion 3: Deepen the Analysis of Regional Disparities in Research Findings

Suggestion 4: Supplement the Visual Representation of Research Results

Suggestion 5: Elaborate on the Research Limitations Regarding the Influence of Socio-economic Factors

Suggestion 6: Enhance the Depth and Breadth of the Discussion Section

Suggestion 7: Clarify the Innovation and Contribution of the Research

Suggestion 8: Standardize the Format of References

Suggestion 9: Increase the Prospectiveness and Recommendations of the Research

Reviewer #2: The manuscript is focused on the problematic of food and nutrition in Australian schools. The manuscript has got a review study. The text is divided into the chapters and subchapters typical for this kind of study. The authors presented very brief introduction, but there are provided all needed kinds of information, which are necessary for the understanding of the study. The text is written in understandable form and it is written by scientific language. The problematic of nutrition, mainly in primary level of education is very important for the development of child from physical and also from mental side. Authors showed all these kinds of information.

The review analysis is very carefully presented. Authors used internationally reputable and established model called PRISMA. The kinds of information from the review study are appropriately added by tables and figures (pictures).

The text seems little bit long and, but these kinds of studies are typical for longer text. I have got only some comments of minor character, which are presented below:

- the first comment has got a technical character, but it is important for the correct version of manuscript. Please read carefully the whole text again and revise some technical mistakes, for example add the explanation of abbreviations under all tables.

- table 3 included quality levels, so please add explanation in the methodology chapter, what is the difference between high, medium and low quality.

To sum up, the text is on high level, after incorporation of comments, it could be prepared for publication purposes.

I hope my comments are helpful

Reviewer #3: Excellent paper.

The authors have described a well-developed methodology in detail.

There is also supported material provided.

It is a well-written and sound paper, nicely presented to the reader.

Useful not only for Australian readers but also to international ones.

**Do you want your identity to be public for this peer review?** For information about this choice, including consent withdrawal, please see our Privacy Policy

Reviewer #1: No

Reviewer #2: No

Reviewer #3: No

---

## [Author Response · Author response to Decision Letter 1]

28 May 2025

We have uploaded new figures as per instructions and the PRISMA Checklist as supplementary materials

---

## [Editor Report · Decision Letter 1]

School food and nutrition environments in Australian primary schools: A scoping review

PONE-D-24-52372R1

Dear Mr/Ms. Gallegos,

We’re pleased to inform you that your manuscript has been judged scientifically suitable for publication and will be formally accepted for publication once it meets all outstanding technical requirements.

Kind regards,

Mc Rollyn Daquiado Vallespin

Academic Editor

PLOS ONE
---

## [Editor Report · Acceptance letter]

PONE-D-24-52372R1

PLOS ONE

Dear Dr. Gallegos,

I'm pleased to inform you that your manuscript has been deemed suitable for publication in PLOS ONE. Congratulations! Your manuscript is now being handed over to our production team.

Kind regards,

on behalf of

Dr. Mc Rollyn Daquiado Vallespin

Academic Editor

PLOS ONE